# Oxidative Stress-Induced TRPV2 Expression Increase Is Involved in Diabetic Cataracts and Apoptosis of Lens Epithelial Cells in a High-Glucose Environment

**DOI:** 10.3390/cells11071196

**Published:** 2022-04-02

**Authors:** Linghui Chen, Yanzhuo Chen, Wen Ding, Tao Zhan, Jie Zhu, Lesha Zhang, Han Wang, Bing Shen, Yong Wang

**Affiliations:** 1Department of Ophthalmology, First Affiliated Hospital of Anhui Medical University, Hefei 230022, China; 1945010724@stu.ahmu.edu.cn (L.C.); 2045011039@stu.ahmu.edu.cn (Y.C.); zhujie@ahmu.edu.cn (J.Z.); 2School of Basic Medical Sciences, Anhui Medical University, Hefei 230032, China; 1945010003@stu.ahmu.edu.cn (W.D.); 1945010006@stu.ahmu.edu.cn (T.Z.); zhanglesha@ahmu.edu.cn (L.Z.); 1914010029@stu.ahmu.edu.cn (H.W.)

**Keywords:** TRPV2, diabetic cataract, apoptosis, human lens epithelial cell

## Abstract

Cataracts are a serious complication of diabetes. In long-term hyperglycemia, intracellular Ca^2+^ concentration ([Ca^2+^]_i_) and reactive oxygen species (ROS) are increased. The apoptosis of lens epithelial cells plays a key role in the development of cataract. We investigated a potential role for transient receptor potential vanilloid 2 (TRPV2) in the development of diabetic cataracts. Immunohistochemical and Western blotting analyses showed that TRPV2 expression levels were significantly increased in the lens epithelial cells of patients with diabetic cataracts as compared with senile cataract, as well as in both a human lens epithelial cell line (HLEpiC) and primary rat lens epithelial cells (RLEpiCs) cultured under high-glucose conditions. The [Ca^2+^]_i_ increase evoked by a TRPV2 channel agonist was significantly enhanced in both HLEpiCs and RLEpiCs cultured in high-glucose media. This enhancement was blocked by the TRPV2 nonspecific inhibitor ruthenium red and by TRPV2-specific small interfering (si)RNA transfection. Culturing HLEpiCs or RLEpiCs for seven days in high glucose significantly increased apoptosis, which was inhibited by TRPV2-specific siRNA transfection. In addition, ROS inhibitor significantly suppressed the ROS-induced increase of TRPV2-mediated Ca^2+^ signal and apoptosis under high-glucose conditions. These findings suggest a mechanism underlying high-glucose–induced apoptosis of lens epithelial cells, and offer a potential target for developing new therapeutic options for diabetes-related cataracts.

## 1. Introduction

Transient receptor potential (TRP) superfamily channels are membrane proteins that act as non-selective cation channels permeable to calcium (Ca^2+^), and play an important role in maintaining Ca^2+^ homeostasis in mammalian tissues and cells [1]. As a member of this superfamily, TRP vanilloid 2 (TRPV2) serves as a sensor for temperature and mechanical stimuli [2,3]. Ca^2+^ can act as a second messenger in cells, and is involved in regulating many different cellular functions, such as the cell cycle, cell proliferation and migration, and cell death [4,5,6]. Ca^2+^ influx mediated by TRPV2 in response to different stimuli can disrupt intracellular Ca^2+^ homeostasis, and Ca^2+^ overload induces a series of catastrophic events in cells. However, the role of TRPV2 in the lens epithelial cells of the eye is still unclear.

The lens is a double convex structure suspended between the anterior chamber and the vitreous of the eye. Its non-vascular transparency helps to refract and focus light onto the retina. Any opacity caused by the lens is typically called a cataract [7]. Diabetes is a chronic metabolic disease that involves multiple systems. Epidemiology studies show that people with diabetes have a higher incidence of cataracts, and the glucose concentration of the crystalline lens can reflect blood glucose levels [8]. In long-term chronic hyperglycemia, the production of reactive oxygen species (ROS) significantly increased, combined with decreased antioxidant capacity, and the Ca^2+^ concentration in the lens is upregulated; [Ca^2+^]_i_ overload induces calpain activation, crystalline proteolysis, lens epithelial cell apoptosis, and a series of other events [9,10]. It is well known that lens epithelial cell apoptosis plays an important role in the development of cataracts [11], but the specific mechanisms involved in Ca^2+^ homeostasis dysfunction and its associated signal transduction in diabetic cataracts remain unclear.

In the present study, we investigated the effect of ROS on TRPV2 expression and function, the functional role of TRPV2 in the regulation of [Ca^2+^]_i_ homeostasis in lens epithelial cells, and the pathologic function of TRPV2 in high glucose-induced lens epithelial cell apoptosis by using, among other approaches, immunohistochemistry, Western blotting, and Ca^2+^ imaging.

## 2. Materials and Methods

### 2.1. Materials

Ruthenium red (RR) and 2-aminoethyldiphenyl borate (2-APB) were obtained from Sigma (St. Louis, MO, USA). Fluo-8/AM was purchased from Abcam (Cambridge, UK), and dissolved in dimethyl sulfoxide. Lipofectamine 3000 was obtained from Invitrogen (Waltham, MA, USA). Specific siRNAs for human TRPV2 (5′-CUUCUUAAACUUCCUGUGUAA-3′) and scrambled siRNA (5′-ACGCGUAACGCGGGAAUUU-3′) were designed by and obtained from Biomics Biotech (Nantong, China). ROS assay kit were obtained from Beyotime Biotechnology (Shanghai, China). Tempol was purchased from MedChemExpress (Monmouth Junction, NJ, USA).

### 2.2. Cell Culture

A human lens epithelial cell (HLEpiC) line was purchased from the American Type Culture Collection (HB-8065, Manassas, VA, USA). HLEpiCs were maintained in Dulbecco’s modified eagle’s medium supplemented with 10% fetal bovine serum (C04001, Bioind, Kibbutz Beit Haemek, Israel) and 1% penicillin/streptomycin (C0222, Beyotime Biotechnology, Shanghai, China) at 37 °C in a 5% CO_2_ incubator. The culture medium in the high-glucose (HG) group contained 25.6 mM glucose, whereas the normal-glucose (NG) culture medium contained 5.5 mM glucose with 20 mM mannitol to maintain the same osmotic pressure as that in the HG condition. Functional studies were performed using cells cultured for 1, 3, 5, and 7 days in the two media. We transfected cells with siTRPV2 (100 nM) and siRNA negative control (siNC, 100 nM) by using Lipofectamine 3000 (L3000015, Thermo Fisher Scientific, Waltham, MA, USA). The transfection efficiency was determined by Western blotting assays.

All animal experiments were conducted in accordance with the approval of the Animal Ethics Committee of Anhui Medical University (protocol code: LLSC20200179). Wild-type Sprague Dawley (SD) rats were humanely killed through the inhalation of CO_2_ gas. After the eyes of rats were washed with sterile phosphate-buffered saline (PBS), the lens capsule membranes were dissected under a dissecting microscope, removed, cut, and transferred to a culture dish containing endothelial cell medium (ECM) (1001, ScienCell Research Laboratories, Carlsbad, CA, USA) complete medium. The membrane tissues were cultured at 37 °C with 5% CO_2_ for 24 h. During seven days of culture, lens epithelial cells gradually emerged from the membrane tissues. Lens epithelial cells, identified by immunofluorescence staining, were used in the experiments, as described below.

### 2.3. [Ca^2+^]_i_ Measurement

HLEpiCs were incubated with 10 μM Fluo-8/AM in an incubator which was kept in the dark for 30 min at 37 °C. Cells were treated with 10 μM RR for 10 min in a normal physiological saline solution containing (in mM) 140 NaCl, 5 KCl, 1 MgCl_2_, 10 glucose, 1 CaCl_2_, and 5 HEPES (pH 7.4). When Ca^2+^ influx reached equilibrium, 250 μM 2-APB was applied for 10 min. Ca^2+^ fluorescence was recorded using fluorescence microscopy (Olympus IX73, Lambda DG-4, Sutter Instrument Company, Novato, CA, USA). We set the excitation wavelength at 490 nm, and the emission wavelength at 525 nm. Changes in the [Ca^2+^]_i_ were expressed as the ratio of fluorescence intensity before and after the 2-APB or RR application (F_1_/F_0_) [12].

### 2.4. RNA Isolation and qRT-PCR

TRIzol reagent (15596-026, Invitrogen, Waltham, MA, USA) was used for RNA extraction from HLEpiCs. The ratio of absorbance at 260 nm and 280 nm was determined to evaluate the quality and concentration of extracted RNA. Single stranded cDNA was synthesized from 1.0 μg of total RNA using the SPARKscript II RT Plus Kit with the gDNA Eraser Kit (AG0304, SparkJade, Qingdao, China).

We used cDNA as a template for qRT-PCR. The reaction system (total volume, 20 μL) was as follows: cDNA template (2 μL), forward and reverse primers (10 μmol/L; 0.8 μL), 2×SYBR qPCR Mix (10 μL) (AH0104, SparkJade), ROX Reference Dye II (0.4 μL) (AH0104, SparkJade) and ddH_2_O (6.8 μL). The reaction conditions were as follows: 94 °C for 3 min; 40 cycles of 94 °C for 10 s, and 60 °C for 30 s. After the reaction, the amplification curve and melting curve were determined. Each sample was assayed three times during amplification of the target gene. The mRNA expression level was evaluated using the 2^−ΔΔCt^ approach, and the relative gene expression level was normalized to actin beta (ACTB). The following primer sequences were used: human TRPV2 (forward CCCGGCTTCACTTCCTCC, reverse GCGTCGGTGTTGGCCTGAC) and human ACTB (forward TCATGAAGTGTGACGTGGACATC, reverse CAGGAGGAGCAATGATCTTGATCT).

### 2.5. Western Blotting

The proteins were extracted with a radioimmunoprecipitation assay buffer (P0013B, Beyotime Biotechnology, Shanghai, China). The total proteins (30 μg) were loaded in each well of a 10% sodium dodecyl sulfate polyacrylamide gel. Subsequently, the proteins were transferred to polyvinylidene fluoride membranes (Millipore, Burlington, MA, USA). After being blocked, the membrane containing the transferred proteins was incubated overnight at 4 °C with the respective following specific primary antibodies: anti-TRPV2 (sc-514848, mouse monoclonal, Santa Cruz Biotechnology, Dallas, TX, USA), anti-Bcl-2 (12789-1-AP, rabbit polyclonal, Proteintech, Wuhan, China), anti-Bax (50599-2-Ig, rabbit polyclonal, Proteintech), anti-caspase-3 (19677-1-AP, rabbit polyclonal, Proteintech), and anti-β-tubulin (AF7011, rabbit polyclonal, Affinity Biosciences, Changzhou, China). The immunosignal resulting after horseradish peroxidase-conjugated secondary antibody incubation was detected using a chemiluminescence detection system. The optical intensity of the protein bands was normalized to β-tubulin, which was run on the same blots, and is presented as relative optical density.

### 2.6. Immunohistochemistry

The Ethics Committee of Anhui Medical University approved the protocol for the use of human tissue (protocol code: P2021-02-10). Briefly, human lens epithelial tissues from patients who were used in our previous study were obtained during clinical surgery, and clinical information is same as our published study [12]. Specimens were collected with written informed consent from each participating patient was obtained. The fresh tissues were fixed with 4% paraformaldehyde, and then the paraffin was embedded and sectioned. The specimens were deparaffinized and rehydrated. Hydrogen peroxide (3%) incubation for 10 min was used to remove endogenous peroxidase activity. Mouse anti-TRPV2 antibodies (1:50) were incubated overnight at 4 °C. After secondary antibody incubation, the specimens were developed with horseradish peroxidase, and then with 3,3′-diaminobenzidine tetrahydrochloride, after which the sections were counterstained with hematoxylin. For the negative control, the primary antibody was omitted. The resulting positive stained cells had brownish or yellow cytoplasm, and were observed and photographed under a light microscope.

### 2.7. TUNEL Assay

The lens epithelial cells were fixed with 4% paraformaldehyde for 20 min, and were permeabilized with 0.2% Triton X-100 at room temperature for 5 min after being rinsed in PBS. After being permeabilized, the cells were labeled with Bright Green Labeling Mix (terminal deoxynucleotidyl transferase and FITC-12-dUTP) for 1 h in the dark. The reaction was stopped by rinsing with PBS three times. The nuclei were visualized using 4′,6-diamidino-2-phenylindole (DAPI). An anti-fluorescence quenching agent was applied, and the resulting cells were observed under a fluorescence microscope. Each cell was observed for blue fluorescence (DAPI) at a wavelength of 460 nm, and the apoptotic cells were identified by green fluorescence at a wavelength of 520 nm. The percentage of apoptotic cells was calculated as follows: (apoptotic cell number/total cell number) × 100.

### 2.8. Immunofluorescence Staining

Briefly, the lens epithelial cells were fixed with 4% paraformaldehyde for 20 min. Subsequently, the membrane was permeabilized by 0.2% Triton X-100, and blocked by 3% bovine serum albumin solution for 1 h. A solution of 5% bovine serum albumin was used as a control. Anti-connexin 46 (YT1048, rabbit polyclonal, ImmunoWay Biotechnology, Suzhou, China) was incubated overnight at 4 °C. After being washed with PBS, the cells were incubated with fluorescent secondary antibody at room temperature for 2 h, rinsed, and mounted with medium containing DAPI to stain cell nuclei for 5 min before an anti-fluorescence quenching agent was applied. The cells were observed under a fluorescence microscope.

### 2.9. ROS Assay

After the treatment in different group, HLEpiCs were loaded with fluorescent probe DCFH-DA diluted at a ratio of 1:1000 in serum-free medium, and cultured at 37 °C with 5% CO_2_ for 30 min. Serum-free cell culture medium was used to wash excess fluorescent dye, and the results were examined by a fluorimeter.

### 2.10. Statistical Analysis

We use two-tailed unpaired Student’s *t*-tests to compare the significance between two groups. Two-way analysis of variance was used when more than two treatments were compared. All analyses were performed with GraphPad Prism. Values are expressed as means ± SEM. A value of *p* < 0.05 was considered statistically significant.

## 3. Results

### 3.1. Changes in TRPV2 Protein Expression

We first used immunohistochemical methods to analyze the expression levels of the TRPV2 channel protein in lens epithelial cells derived from diabetic cataract tissue compared with those derived from senile cataract tissue as control group. The results showed that the expression level of TRPV2 protein was significantly higher in the diabetic cataract tissue (Figure 1a,b). We then cultured HLEpiCs and primary cultured rat lens epithelial cells in a high-glucose (25.6 mM) medium to mimic a high-glucose environment for 1, 3, 5, and 7 days. The primary cultured rat lens epithelial cells were verified by assessing the expression of connexin-46 (Figure 1c). Western blotting analysis showed that, compared with NG culture, the expression levels of TRPV2 protein in the two types of cells cultured in HG were significantly increased in the seventh day (Figure 1d–g). Given that the TRPV2 channel protein is permeable to Ca^2+^, we speculated that a Ca^2+^ imbalance may play a role in the occurrence and development of diabetic cataracts.

### 3.2. Changes of TRPV2-Mediated Ca^2+^ Influx in an HG Environment

HLEpiCs and primary cultured rat lens epithelial cells were each cultured in NG or HG medium for seven days. The Ca^2+^ imaging results firstly showed that high glucose treatment for seven days significantly increased basal intracellular Ca^2+^ concentration in HLEpiCs (Appendix A). Then, Ca^2+^ influx in HLEpiCs was evoked by the non-specific TRPV2 channel agonist 2-APB [13,14], but was inhibited by the non-specific TRPV2 channel inhibitor RR (Figure 2a–f) [15,16] in a normal physiological saline solution. Moreover, we found that a 2-APB-induced [Ca^2+^]_i_ increase was enhanced in HLEpiCs and primary rat lens epithelial cells cultured under HG compared with cells cultured in NG (Figure 2a–f). To further explore the role of TRPV2 in 2-APB-evoked [Ca^2+^]_i_ increase, HLEpiCs were cultured in NG or HG medium for seven days, and then transfected with TRPV2 siRNA or with scrambled siRNA as a control. Our Ca^2+^ imaging results showed that 2-APB-induced [Ca^2+^]_i_ increase was significantly reduced in TRPV2 siRNA-transfected HLEpiCs compared with scrambled siRNA-transfected HLEpiCs, both in the NG and HG environments (Figure 2j,k). In addition, TRPV2 siRNA transfection successfully suppressed TRPV2 expression in HLEpiCs (Figure 2g–i). To further investigate whether 2-APB-induced [Ca^2+^]_i_ increase was due to Ca^2+^ influx or endogenous Ca^2+^ release, HLEpiCs were stimulated with 2-APB in a Ca^2+^-free solution in both NG and HG groups. Our results showed that, in the Ca^2+^ free solution, 2-APB induced an increase in [Ca^2+^]_i_ in HLEpiCs and followed Ca^2+^ influx evoked through the application of 1 mM extracellular Ca^2+^. Moreover, 2-APB-induced Ca^2+^ release was increased in the HG group even there was no significant difference, but 2-APB-induced Ca^2+^ influx was significantly increased in the HG group compared with the NG group (Appendix A). In another group, HLEpiCs were treated with BTP2 (a nonselective inhibitor of Orai channels) to inhibit store-operated Ca^2+^ entry (SOCE). Our results suggested that BTP2 treatment suppressed 2-APB-induced Ca^2+^ releases in the Ca^2+^ free solution, even there were no significant differences in HLEpiCs cultured under NG and HG conditions for seven days, but significantly suppressed Ca^2+^ influx in HLEpiCs cultured under HG conditions for seven days. Interesting, 2-APB-induced Ca^2+^ influx in BTP2-treated group was still significantly increased in the HG group compared with the NG group (Appendix A). Therefore, 2-APB-induced Ca^2+^ influx was significantly stronger in the HG group both with and without BTP2 treatment. These findings suggest that TRPV2 is involved in mediating Ca^2+^ influx in lens epithelial cells, and that this effect is enhanced in an HG environment, which may lead to [Ca^2+^]_i_ overload.

### 3.3. Role of TRPV2 in HG-Induced Apoptosis of Lens Epithelial Cells

Abnormal apoptosis of lens epithelial cells is the initial event in cataract development, and Ca^2+^ is involved in regulating apoptosis. Therefore, we investigated the apoptotic changes of lens epithelial cells under HG conditions. HLEpiCs and primary cultured rat lens epithelial cells were each cultured in NG or HG medium for seven days. In the HG group, Western blotting analysis of apoptosis-related proteins showed that the intracellular Bcl-2/Bax ratio was significantly decreased, whereas the cleaved caspase-3 protein expression level was significantly increased (Figure 3a–f). In addition, TUNEL analysis showed an increased percentage of apoptotic cells (Figure 3g–j). These data suggest that apoptosis of lens epithelial cells is increased under HG conditions.

To further assess the role of TRPV2 in HG-induced apoptosis, we used Western blotting to analyze apoptosis-related proteins. In the NG group, transfection with TRPV2 siRNA did not significantly alter the Bcl-2/Bax ratio, cleaved caspase-3 protein expression levels (Figure 4a–c), or cleaved caspase-9 protein expression levels (Appendix A), compared with scrambled siRNA transfection in HLEpiCs. In the HG group, transfection with TRPV2 siRNA increased the proportion of Bcl-2/Bax in cells, whereas cleaved caspase-3 (Figure 4d–f) and cleaved caspase-9 (Appendix A) protein expression levels were significantly decreased. These data indicate that an HG environment significantly stimulates apoptosis of HLEpiCs, and that TRPV2 may play a key role in this apoptosis.

### 3.4. Role of ROS in TRPV2 Expression Regulation

A high-glucose environment is able to induce excessive ROS production [17]. Our results also showed that ROS was significantly increased in HLEpiCs cultured in HG media, but inhibited by the antioxidant Tempol (Figure 5a: 0.25, 0.5, 1 mM in three days group; Figure 5b: 0.5, 1 mM in seven days group) treatment for three and seven days with a concentration-dependent manner. Then, we cultured HLEpiCs in NG or HG medium for seven days, and the cells were treated with different concentrations of Tempol (0, 0.25, 0.5, 1, 2 mM). In the HG group, Western blotting analysis showed that, compared with the solvent control, the increased expression levels of TRPV2 protein and cleaved caspase-3 protein were significantly suppressed, whereas Bcl-2/Bax ratio was significantly increased in HLEpiCs (Figure 5c–f). In addition, the Ca^2+^ imaging results showed that 2-APB-induced [Ca^2+^]_i_ increase was significantly inhibited in HLEpiCs treated with Tempol (1 mM) for seven days in both NG and HG groups (Figure 5g,h). To further investigate the effect of ROS on TRPV2, mitochondrial membrane potential was measured. Our results showed that the membrane potential of HLEpiCs was significantly decreased in HG medium culture for seven days, which was partially reversed by the transfection of TRPV2 siRNA or the treatment of cells with Tempol (1 mM) (Appendix A). In a high-glucose environment, increased apoptosis may lead to the loss of mitochondrial membrane potential, and ROS-induced increase of TRPV2 expression may be involved in this process.

## 4. Discussion

In the present study, we first found elevated expression of the TRPV2 channel protein, which is primarily permeable to Ca^2+^, in the lens epithelium of patients with diabetic cataracts, compared with that among patients with senile cataracts. We then cultured lens epithelial cells in an HG medium to simulate in vitro diabetes. The expression levels of TRPV2 in both HLEpiCs and primary cultured rat lens epithelial cells were increased under HG conditions. A normal glucose culture medium (5.5 mM) with mannitol (20 mM) was used as a control group, therefore HG-induced changes are not due to the effect of osmolarity change. In an experiment measuring [Ca^2+^]_i_, our results showed that the Ca^2+^ influx induced by the non-specific TRPV2 channel agonist 2-APB was significantly enhanced in HG cultured lens epithelial cells, and that this Ca^2+^ influx could be inhibited by the non-specific TRPV2 channel inhibitor RR. To further assess the role of TRPV2 in this enhanced Ca^2+^ influx, we used specific siRNA to inhibit TRPV2 expression. Transfection of TRPV2 siRNA significantly inhibited TRPV2 expression and 2-APB–induced Ca^2+^ influx under both NG and HG conditions, but the inhibition was stronger in the HG medium. In addition, the HG culture significantly enhanced ROS production, and the ROS inhibitor significantly suppressed ROS-induced increase of TRPV2-mediated Ca^2+^ signal and apoptosis under high-glucose conditions. These findings suggest that an HG environment stimulates ROS production, which enhances TRPV2 expression. High expression of TRPV2 in lens epithelial cells, and the resulting enhanced Ca^2+^ influx, leads to intracellular Ca^2+^ homeostasis and signal transduction abnormalities and Ca^2+^ overload. Eventually, the apoptosis of lens epithelial cells is increased, and may be related to the development of diabetes-related cataracts.

Cataracts are the leading cause of blindness in the world, and people with diabetes are at higher risk of both cataract occurrence and more rapid progression [8]. Previous studies have suggested that the polyol pathway is the main explanation for the formation of diabetic cataracts, and other mechanisms, including oxidative stress and glycosylation, have also been suggested [17,18,19]. The glucose concentration of the lens in a diabetes environment does not depend on insulin, and glucose in the lens is reduced to sorbitol, which is difficult to convert into fructose [20,21]. The resulting hypertonic environment induces apoptosis of lens epithelial cells, and causes water to enter the lens, leading to continuous swelling. It has been reported that the Ca^2+^ concentration in the diabetic lens is increased, and that Ca^2+^ antagonists exhibit anticataract properties [10,12,22,23]. Our results also showed that TRPV2 was markedly increased in HG-cultured lens epithelial cells. Moreover, Ca^2+^ influx mediated by TRPV2 is involved in HG-induced apoptosis of lens epithelial cells. Although the HG cell culture model is not equivalent to the diabetic condition, cultures can mimic the HG environment found in vivo. In addition, our immunohistochemical results showed that the expression level of TRPV2 in the lens epithelial cells of patients with diabetic cataracts was increased. Therefore, our findings suggest that TRPV2 may play a role in the occurrence and development of diabetic cataracts. In our previous study, we identified that Orai3 as a SOCE channel mediates the increase of [Ca^2+^]_i_ in lens epithelial cells through the SOCE pathway in a high-glucose environment [12]. However, in the present study, after an Orai channel-mediated Ca^2+^ influx was suppressed by a SOCE inhibitor, we still recorded an increased TRPV2-mediated Ca^2+^ influx in the HG environment. Thus, besides Orai3, TRPV2 may also involve in HG-induced Ca^2+^ overload in lens epithelial cells.

Previous studies have shown that TRPV2 has important roles in the proliferation, invasion, and apoptosis of tumor cells [24,25,26,27]. For example, it has been reported that inhibition of TRPV2 expression reduces the proliferation and invasion of esophageal squamous cell carcinoma cells [25]. In urothelial carcinoma and endometrial carcinoma, TRPV2 is highly expressed in cancer tissues, and the TRPV2 agonist cannabidiol activates apoptosis [28,29]. Several studies in the past few years have also identified a role for TRPV2 in cardiac function, and have suggested that inhibition of TRPV2 activity may be a potential target for the treatment of cardiomyopathy and heart failure [30,31,32,33]. However, a role for TRPV2 in lens epithelial cells has not been previously reported. In our study, we found that the apoptosis of lens epithelial cells was significantly increased under HG conditions, and that TRPV2 may be involved in this process. Although the Bcl-2/Bax protein expression ratio was significantly decreased in lens epithelial cells cultured in the HG medium, the expression level of active caspase-3 was significantly increased. To further evaluate the role of TRPV2 in lens epithelial cell apoptosis, we transfected a specific TRPV2 siRNA into lens epithelial cells to inhibit TRPV2 expression. Our results showed that TRPV2 siRNA transfection significantly increased the ratio of Bcl-2/Bax, and significantly suppressed cleaved caspase-3 protein expression level in HLEpiCs cultured in HG, but not in NG. This suggests that the TRPV2 channel protein may play a critical role in the apoptosis of lens epithelial cells that is substantially increased in an HG environment. Therefore, our study suggests a potential target for the treatment of diabetic cataracts, although in vivo animal model studies will be needed to evaluate the therapeutic value of TRPV2 in diabetic cataract.

Increased blood glucose in diabetes induces oxidative stress in cells, and ROS has been reported to increase the expression of TRPV2, which mediates cell death in human hepatoma cells [17,34]. Our results showed that intracellular ROS and TRPV2 proteins were significantly increased under high glucose environment. To verify the potential relationship between ROS and TRPV2, we treated the cells with an antioxidant Tempol, and found that when ROS was inhibited, HG-enhanced TRPV2 expression was suppressed as well. These data indicated that TRPV2 may be a ROS-regulated channel, and our findings were similar to several studies in other groups [16,34]. In addition, the cell apoptosis induced by HG culture was also attenuated. Therefore, these results may indicate that increased ROS in high-glucose environments may induce TRPV2 expression enhancement, thereby enhancing Ca^2+^ overload, and following cell apoptosis in lens epithelial cells. TRPV2 may be a potential target in high-glucose environment-induced lens epithelial cell apoptosis and diabetic cataracts. However, in the present study, we also found that under NG-cultured conditions, ROS and TRPV2 were both not significantly suppressed by antioxidant, but 2-APB-induced Ca^2+^ increase was dramatically reduced. The phenomenon may be caused by some TRPV2 activation component suppression in Tempol treatment, but further study is needed in the future.

## 5. Conclusions

In conclusion, our study showed that TRPV2 expression was significantly increased in the epithelial lens tissue of patients with diabetic cataracts, and that enhanced ROS induced lens epithelial cell apoptosis by upregulating TRPV2 expression and TRPV2-mediated Ca^2+^ overload in an HG environment, suggesting that TRPV2 may be one of key ion channels mediating Ca^2+^ influx in lens epithelial cells, and may provide a potential therapeutic target for diabetic or HG-induced cataracts.

## Figures and Tables

**Figure 1 cells-11-01196-f001:**
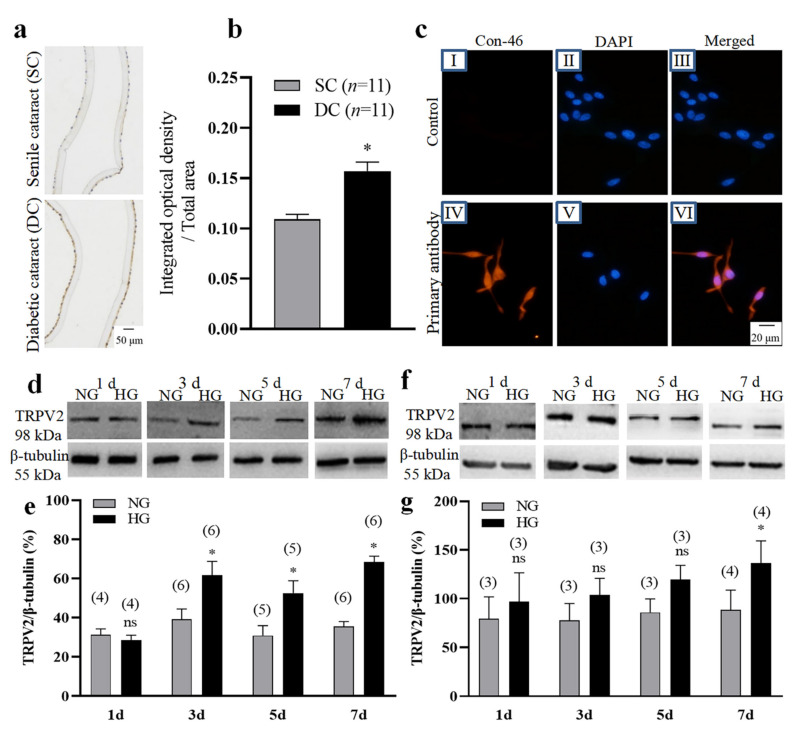
Changes in TRPV2 protein expression. (**a**,**b**) Representative images (**a**) and summary data (**b**) showing TRPV2 expression levels in human lens epithelial cells from patients having senile cataracts (SC) or diabetic cataracts (DC). In (**b**), data showing the ratio of integrated optical density/total area of TRPV2. (**c**) Immunofluorescence images showing the expression of connexin-46 (Con-46) protein in primary cultured lens epithelial cells from rat. No—primary antibody control: I–III Connexin-46 expression: IV-VI. Cell nuclei are stained blue by 4′,6-diamidino-2-phenylindole (DAPI). (**d**–**g**) Representative Western blotting images (**d**,**f**) and summary data (**e**,**g**) showing the expression levels of TRPV2 in HLEpiCs (**d**,**e**) and primary rat lens epithelial cells (**f**,**g**) cultured in a normal-glucose (NG, 5.5 mM glucose and 20 mM mannitol) or high-glucose (HG, 25.6 mM glucose) media for 1, 3, 5, and 7 days. Numbers in parentheses over the bars in panels (**e**,**g**) represent the number of biological replicates. Values are shown as the mean ± SEM; *n* = 3–11. * *p* < 0.05 vs. SC or NG at 3, 5, and 7 days analyzed by two-tailed unpaired Student’s *t* test; ns, not significant.

**Figure 2 cells-11-01196-f002:**
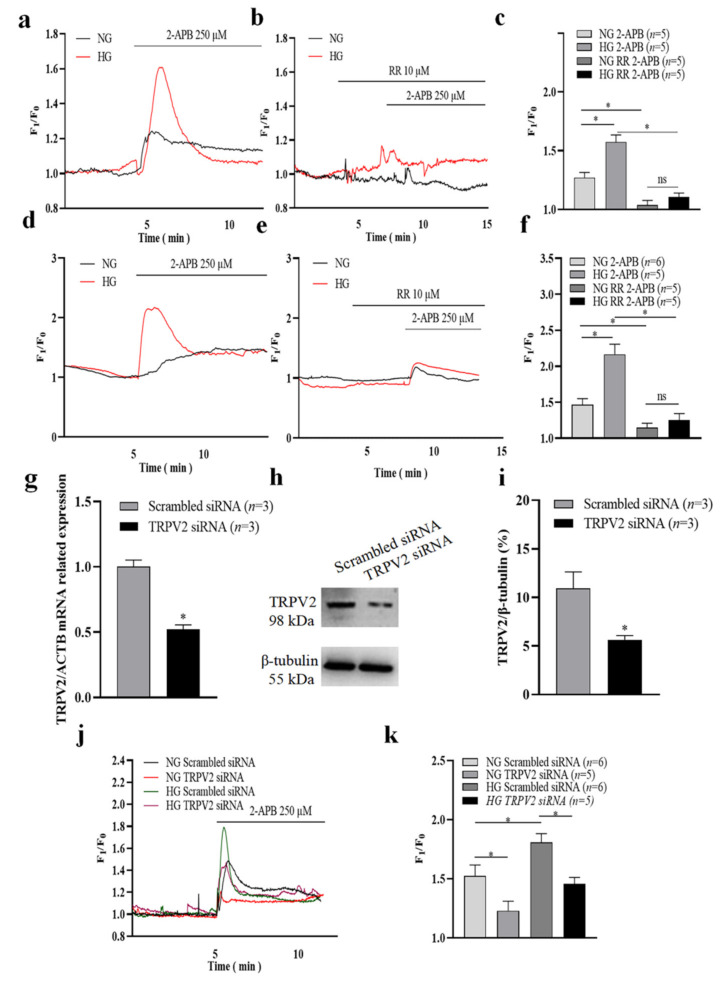
Changes of TRPV2-mediated Ca^2+^ influx in a high-glucose environment. (**a**–**f**) Representative traces (**a**,**b**,**d**,**e**) and summary data (**c**,**f**) showing the changes in the intracellular Ca^2+^ concentration of HLEpiCs (**a**–**c**) and primary rat lens epithelial cells (**d**–**f**) cultured in a normal-glucose (NG, 5.5 mM glucose and 20 mM mannitol) or high-glucose (HG, 25.6 mM glucose) medium for seven days. The cells were treated without or with ruthenium red (RR, 10 μM) and activated by 2-aminoethyldiphenyl borate (2-APB, 250 μM). (**g**) Summary data showing TRPV2 mRNA expression level of HLEpiCs transfected with TRPV2 siRNA or scrambled siRNA control for three days. (**h**–**i**) Representative Western blotting images (**h**) and summary data (**i**) showing TRPV2 protein expression levels of HLEpiCs transfected with TRPV2 siRNA or scrambled siRNA control for three days. (**j**–**k**) Representative traces (**j**) and summary data (**k**) of the changes in intracellular Ca^2+^ concentration of HLEpiCs transfected with TRPV2 siRNA or scrambled siRNA control and cultured in NG or HG media for seven days. The TRPV2 channel agonist 2-APB (250 μM) activated the channel to induce Ca^2+^ influx. Values are shown as the mean ± SEM; *n* = 3–6. * *p* < 0.05 analyzed by two-way analysis of variance followed by a Bonferroni test in panel **c**, **f** and **k**, and two-tailed unpaired Student’s *t* test in panel (**g**,**i**); ns, not significant.

**Figure 3 cells-11-01196-f003:**
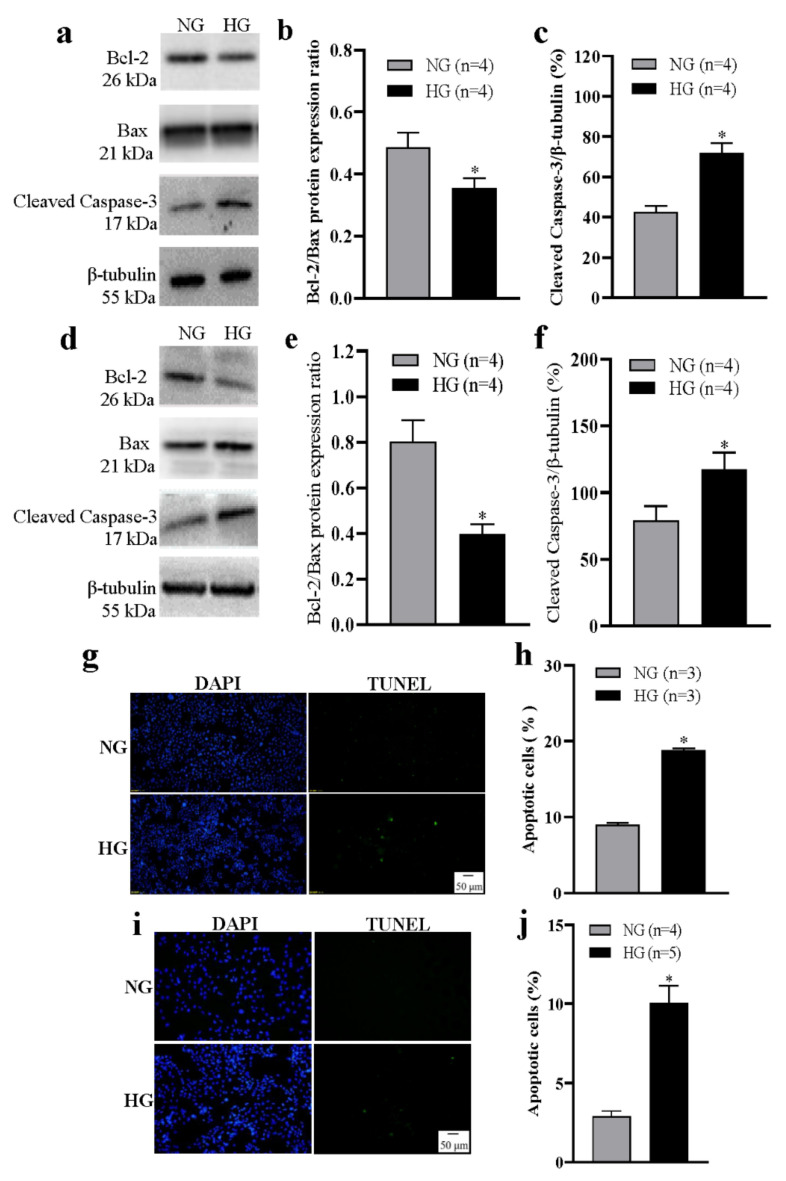
High glucose–induced apoptosis of lens epithelial cells. (**a**–**f**) Representative Western blotting images (**a**,**d**) and summary data (**b**,**c**,**e**,**f**) showing the Bcl-2/Bax protein expression ratio (**b**,**e**) and the expression levels of active caspase-3 (**c**,**f**) in HLEpiCs (**a**–**c**) and primary rat lens epithelial cells (**d**–**f**) cultured in a normal-glucose (NG, 5.5 mM glucose and 20 mM mannitol) or high-glucose (HG, 25.6 mM glucose) medium for seven days. (**g**,**i**) Representative images showing cell nuclei (blue, 4′,6-diamidino-2-phenylindole [DAPI]) and apoptotic HLEpiCs (**g**) and primary rat lens epithelial cells (**i**) (green, terminal deoxynucleotidyl transferase–mediated nick-end labeling) cultured in NG or HG media for seven days. (**h**,**j**) Summary data showing the percentage of apoptotic HLEpiCs (**h**) and primary rat lens epithelial cells (**j**). Values are shown as the mean ± SEM; *n* = 3–5. * *p* < 0.05 vs. NG analyzed by a two-tailed unpaired Student’s *t* test.

**Figure 4 cells-11-01196-f004:**
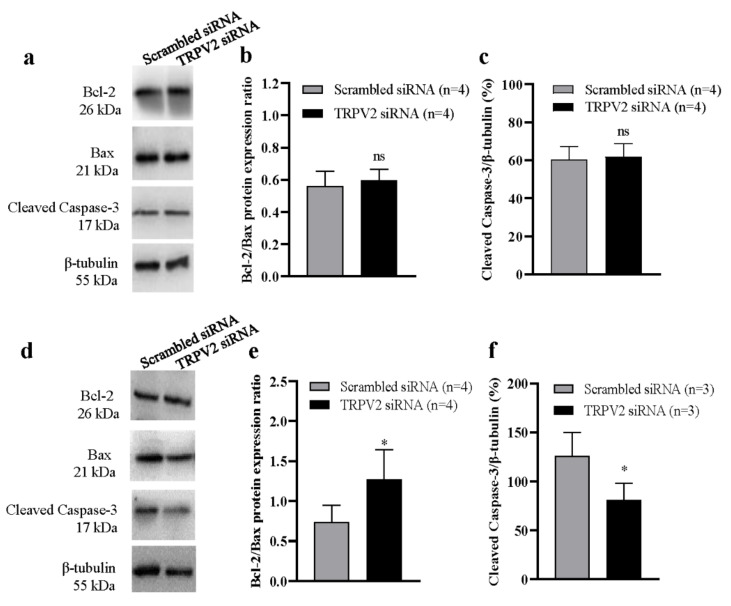
Role of TRPV2 in high glucose–induced apoptosis of lens epithelial cells. Representative Western blotting images (**a**,**d**) and summary data (**b**,**c**,**e**,**f**) showing Bcl-2/Bax protein expression ratios and the levels of active caspase-3 in HLEpiCs cultured in a normal-glucose ((**a**–**c**), 5.5 mM glucose and 20 mM mannitol) or high-glucose ((**e**,**f**), 25.6 mM glucose) medium for seven days and transfected with TRPV2 siRNA or scrambled siRNA control. Values are shown as the mean ± SEM; *n* = 3–4. * *p* < 0.05 vs. scrambled siRNA analyzed by two-tailed unpaired Student’s *t* test; ns, not significant.

**Figure 5 cells-11-01196-f005:**
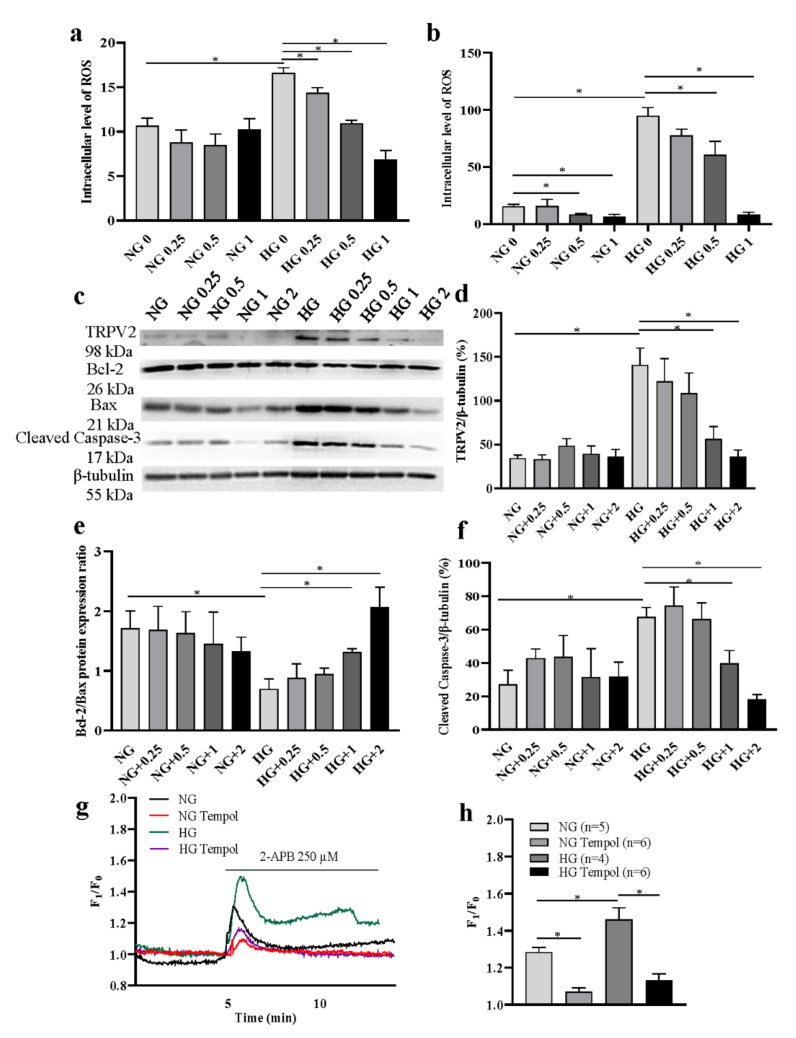
Effects of reactive oxygen species (ROS) on TRPV2 expression and function. (**a**,**b**) Summary data showing intracellular ROS level in HLEpiCs treated with different concentrations of an antioxidant Tempol (0, 0.25, 0.5, 1 mM) for three and seven days (*n* = 4). (**c**–**f**) Representative Western blotting images (**c**) and summary data (**d**–**f**) showing the expression levels of TRPV2 (*n* = 4) (**d**), Bcl-2/Bax (*n* = 3) protein expression ratio (**e**) and active caspase-3 in HLEpiCs (**f**) cultured in a normal-glucose (NG, 5.5 mM glucose and 20 mM mannitol) or high-glucose (HG, 25.6 mM glucose) media treated with Tempol (0, 0.25, 0.5, 1, 2 mM). (**g**,**h**) Representative traces (**g**) and summary data (**h**) showing the changes in the intracellular Ca^2+^ concentration of HLEpiCs treated with Tempol (1 mM) cultured in a NG or HG medium for seven days. The cells were activated by 2-aminoethyldiphenyl borate (2-APB, 250 μM). Values are shown as the mean ± SEM; *n* = 3–6. * *p* < 0.05 analyzed by two-way analysis of variance, followed by a Bonferroni test.

## Data Availability

Not applicable.

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
