# Peer review of "Oxidative Stress-Induced TRPV2 Expression Increase Is Involved in Diabetic Cataracts and Apoptosis of Lens Epithelial Cells in a High-Glucose Environment"

_cells, 2022, doi:10.3390/cells11071196_

Round 1
Reviewer 1 Report
The present manuscript investigated the consequence of high glucose environment for the lens epithelial cells and the development of cataract associated with diabetes. The authors reported an increased expression of the non-selective ion channel TRPV2 in cultures maintained in high glucose for 7 days. Functionally, TRPV2 was stimulated with 2-APB and inhibited with ruthenium red (RR), both nonspecific compounds. The increase Ca2+ response in high glucose (HG) condition was associated with increased cell apoptosis. In addition, siRNA targeting TRPV2 was used as a strategy to firmly establish the role of TRPV2 in both Ca2+ response to 2-APB and the increase apoptosis. The increased TRPV2 expression was causally linked with enhanced ROS production consecutive to high glucose.
The design of the study is straightforward, and the conclusion raised by the authors are mainly supported by the experiments.
Here are some specific points:
The increase expression of TRPV2 in HG could lead to a Ca2+ overload as suggested by the authors. That implies however that the channels are opened in resting condition. The experiments using RR to block TRPV2 do not show a sign of elevated basal Ca2+ concentration (no decrease of the resting Ca2+ level upon RR addition, fig 2b and 2e). The question is thus, what is the basal Ca2+ level in NG (normal glucose) and in HG condition? Is there any sign of Ca2+ overload in cells cultured in high glucose medium? As all data are normalized, one cannot evaluate this important parameter.
As a control, it would be worth repeating the 2-APB stimulation in a medium lacking Ca2+, to confirm the 2-APB-induced Ca2+ entry. Especially considering the lack of 2-APB specificity.
In a recent letter (Wang et al. 2021; Clinical and translational medicine) the same group reported and enhancement of Orai3 (and STIM1) expression that would lead to Ca2+ overload and participate to the pathogenic effect of high glucose induced cataract. Strikingly, this previous result is not discussed in the present study and that has to be done. In particular as 2-APB is a known opener of Orai3 channel (e.g. Kappel et al., 2020 Int J Mol Sci). What is the current hypothesis regarding these two channels as potential major players in the dysregulated Ca2+ homeostasis of lens epithelial cells?
Others:
References from the literature should be provided for the use of RR and 2-APB as modulator of TRPV2 channel activity.
Fig 2j-k: on the traces (j) there is no difference between NG scramble siRNA and NG TRPV2 siRNA, while on the statistics (k) there is a 50% decrease. Either the trace is not representative, or the statistics are wrong. Please correct.
p.4 section 3.1:” TRPV2 channel protein is mainly permeable to Ca2+… “ it is not the case, TRPV2 is reported as a non-selective cation channel (e.g. Caterina et al., Nature 1999)
Materiel &Methods :
Section 2.3: the fluorescence microscope should be described as well as the filters used for calcium imaging.
The external solution contains 1 mM CaCl2 and 0.2 EGTA, how much free calcium is that?
Author Response
Point-to-point Response
General response: We kindly thank reviewer for your constructive suggestions. In response to your comments, we have followed your suggestions to do additional experiments and revised the manuscript carefully. We believe that this version of the manuscript is largely improved.
#1 Reviewer:
Comment 1: The increase expression of TRPV2 in HG could lead to a Ca2+ overload as suggested by the authors. That implies however that the channels are opened in resting condition. The experiments using RR to block TRPV2 do not show a sign of elevated basal Ca2+ concentration (no decrease of the resting Ca2+ level upon RR addition, fig 2b and 2e). The question is thus, what is the basal Ca2+ level in NG (normal glucose) and in HG condition? Is there any sign of Ca2+ overload in cells cultured in high glucose medium? As all data are normalized, one cannot evaluate this important parameter.
Answer: Thank you very much for your comments. To respond to this issue, we performed additional experiments using the radiometric dye Fura-2 to measure the absolute value of intracellular Ca2+ concentration. Our data showed that high glucose treatment for 7 days significantly increased basal intracellular Ca2+ concentration in HLEpiCs (Rebuttal Figure 1), which may be mediated by TRPV2 upregulation in high glucose environments. We provided the data as a supplemental data for readers.
Method: HLEpiCs were incubated with Fura-2/AM (2 μM, Abcam, ab120873) and 0.02% Pluronic F-127 (Sigma-Aldrich, P2443-250G) in an incubator kept in the dark for 30 min at 37 °C. Basal intracellular Ca2+ concentration were measured for 10 min in a normal physiological saline solution containing (in mM) 140 NaCl, 5 KCl, 1 MgCl2, 10 glucose, 1 CaCl2, and 5 HEPES (pH 7.4). Ca2+ fluorescence was recorded using fluorescence microscopy. Changes in the [Ca2+]i were expressed as the ratio of 340/380.
Rebuttal Figure 1
Comment 2: As a control, it would be worth repeating the 2-APB stimulation in a medium lacking Ca2+, to confirm the 2-APB-induced Ca2+ entry. Especially considering the lack of 2-APB specificity.
Answer: Thank you very much for your comments. To respond to this issue, we performed additional experiments. In the study, after HLEpiCs was cultured under NG or HG media for 7 days, cells were stimulated with 2-APB in Ca2+-free solution containing (in mM) 140 NaCl, 5 KCl, 1 MgCl2, 10 glucose, 0.2 EGTA, and 5 HEPES (pH 7.4). Our results showed that in Ca2+ free solution, 2-APB (250 μM) induced an increase in intracellular Ca2+ concentration ([Ca2+]i) in HLEpiCs, and followed Ca2+ influx was evoked by application of 1 mM extracellular Ca2+. The increase of [Ca2+]i was more obvious in high glucose group whatever intracellular Ca2+ release or extracellular Ca2+ influx (Rebuttal Figure 2). This suggests that 2-APB-induced increase of [Ca2+]i is realized through intracellular Ca2+ release and extracellular Ca2+ influx. In Figure 2a-c, ruthenium red (RR, 10 μM) almost abolished 2-APB-induced rise of [Ca2+]i in Ca2+-containing normal physiological saline solution. Therefore, we think that TRPV2 should involve in Ca2+ influx.
Rebuttal Figure 2
Comment 3: In a recent letter (Wang et al. 2021; Clinical and translational medicine) the same group reported and enhancement of Orai3 (and STIM1) expression that would lead to Ca2+ overload and participate to the pathogenic effect of high glucose induced cataract. Strikingly, this previous result is not discussed in the present study and that has to be done. In particular as 2-APB is a known opener of Orai3 channel (e.g. Kappel et al., 2020 Int J Mol Sci). What is the current hypothesis regarding these two channels as potential major players in the dysregulated Ca2+ homeostasis of lens epithelial cells?
Answer: Thank you very much for your comments. It is yes that we did not discuss this issue and we agree your viewpoint to clarify the involvement of Orai3 and TRPV2 in high glucose induced cataract. To answer this question, we did additional intracellular Ca2+ concentration ([Ca2+]i) measurement experiments. BTP2 is a nonselective inhibitor of Orai channels which have three types including Orai1, Orai2 and Orai3 mediating store-operated Ca2+ entry (SOCE) [1,2]. In the study, we used BTP2 to inhibit Orai channels. Our results suggested that BTP2 (10 μM) treatment in HLEpiCs cultured under NG or HG conditions for 7 days significantly suppressed 2-APB-induced [Ca2+]i rise, however did not completely inhibit 2-APB-induced [Ca2+]i rise. More importantly, 2-APB-induced [Ca2+]i rise was still significantly stronger in HG group whatever with or without BTP-2 treatment (Rebuttal Figure 3). These results indicated that maybe Orai channel and TRPV2 channel are both involved in high glucose-induced abnormal of Ca2+ homeostasis of lens epithelial cells. We added these data into supplemental data and discussed this issue in this revised version of manuscript.
Rebuttal Figure 3
Reference:
1.D'Souza, R.S.; Lim, J.Y.; Turgut, A.; Servage, K.; Zhang, J.; Orth, K.; Sosale, N.G.; Lazzara, M.J.; Allegood, J.; Casanova, J.E. Calcium-stimulated disassembly of focal adhesions mediated by an ORP3/IQSec1 complex. Elife. 2020, 9, 54113; DOI: 10.7554/eLife.54113.
2.Xu, F.; Wang, Y.; Gao, H.; Zhang, X.; Hu, Y.; Han, T.; Shen, B.; Zhang, L.; Wu, Q. X-Ray Causes mRNA Transcripts Change to Enhance Orai2-Mediated Ca2+ Influx in Rat Brain Microvascular Endothelial Cells. Front Mol Biosci. 2021, 8, 646730; DOI: 10.3389/fmolb.2021.646730.PMID: 34595206
Comment 4: Others:
- References from the literature should be provided for the use of RR and 2-APB as modulator of TRPV2 channel activity.
Answer: Thank you very much for your suggestions. We have added relevant references, please check the manuscript of this version.
- Fig 2j-k: on the traces (j) there is no difference between NG scramble siRNA and NG TRPV2 siRNA, while on the statistics (k) there is a 50% decrease. Either the trace is not representative, or the statistics are wrong. Please correct.
Answer: Thank you very much for your careful check. This is our mistake to show an un-representative trace in Figure 2j. We corrected it. Please check this revised Figure 2j.
- 4 section 3.1:” TRPV2 channel protein is mainly permeable to Ca2+… “ it is not the case, TRPV2 is reported as a non-selective cation channel (e.g. Caterina et al., Nature 1999)
Answer: Thank you very much for your suggestion. Yes, we revised this sentence and delete the word “mainly”. TRPV2 is usually described as a non-selective cation channel, but it has been reported that it mediates cation currents in the order of osmosis is Ca2+ > Mg2+ > Na+ ~ Cs+ > K+. The relative permeability ratio of PCa2+/PNa+ is 2.94 [1,2]. Therefore, many current studies focus on the Ca2+ permeability of TRPV2. Please check the sentence again.
Reference:
- Kojima, I.; Nagasawa, M. TRPV2. Handb Exp Pharmacol. 2014, 222, 247-72; DOI: 10.1007/978-3-642-54215-2_10.
- Shibasaki, K. Physiological significance of TRPV2 as a mechanosensor, thermosensor and lipid sensor. J Physiol Sci. 2016, 66, 359-65; DOI: 10.1007/s12576-016-0434-7.
Comment 5: Materiel & Methods :
- Section 2.3: the fluorescence microscope should be described as well as the filters used for calcium imaging.
Answer: Thank you very much for your advice. We have added this information, please check this version of the manuscript.
- The external solution contains 1 mM CaCl2 and 0.2 EGTA, how much free calcium is that?
Answer: Thank you very much for your question. Sorry for our mistake. Actually, EGTA was not used in this experiment, and the free calcium concentration was 1 mM. We corrected this mistake. In some Ca2+-free solution experiment in question 2, we used 0.2 mM EGTA to chelate the extracellular Ca2+. Please check this revision of the manuscript.

Reviewer 2 Report
In this manuscript, Chen et al. showed that the exposure of high glucose for 7 days to human and rat lens epithelial cells induced Ca2+ overload following by up-regulation of TRPV2, resulting in ROS-mediated cell apoptosis. It has been reported that hyperglycemia potentiated TRPV2 activity in non-lens epithelial cells. This manuscript includes novel findings; however, there are some concerns that need to be addressed.
Major concerns:
- Drug application should be expressed by the bar but not the arrow. 2-APB continuously keeps [Ca2+]i Was 2-APB treated acutely in Figure 2?
- The authors should describe/explain that HG-induced changes are not due to the changes in osmolarity.
- In Figure 1, the authors showed the increased expression of TRPV2 proteins in HG group. The authors indicate which TRPV subtypes are expressed in mammalian lens epithelial cells and if their expressions are changes under HG treatments.
- The authors should clarify the mechanisms underlying increased expression of TRPV2 proteins by HG stimulation. Decreased TRPV2 protein degradation or increased TRPV2 transcription, etc.
- In Figure 5, the authors suggested that HG-mediated (for 7 days) increase in ROS production might cause increased TRPV2 protein levels. To determine whether TRPV2 is upstream of ROS-dependent Bax/Bcl2 and caspase pathway, the authors showed the time-dependent changes in the TRPV2 and Bax/Bcl2 levels in Figure 3 and 5. Without the data, the reviewer cannot agree with the authors’ conclusion.
- In cell models but not animal models of hyperglycemia (high glucose exposure), ion channel expressions are recovered after one week. Please explain the reasons why the HG-mediated increase in TRPV2 protein expression is maintained for over 7 days.
- In Figure 4, Ca2+ imaging experiments should be performed using TRPV2-knockdown cells.
- English editing is needed.
Author Response
Point-to-point Response
General response: We kindly thank reviewer for your constructive suggestions. In response to your comments, we have followed your suggestions to revise the manuscript carefully. We believe that this version of the manuscript is largely improved.
#2 Reviewer:
Comment 1: Drug application should be expressed by the bar but not the arrow. 2-APB continuously keeps [Ca2+]i was 2-APB treated acutely in Figure 2?
Answer: Thank you very much for your suggestions. We added the bar to indicate the drug application in this revision. 2-APB was continuously to maintain [Ca2+]i. Please check this revised version of the manuscript.
Comment 2: The authors should describe/explain that HG-induced changes are not due to the changes in osmolarity.
Answer: Thank you very much for your advice. In our study, normal glucose medium (5.5 mM) with mannitol (20 mM) was used to balance osmotic culture cells as control group. We have described in the manuscript that the changes induced by high glucose are not due to the changes in osmotic pressure. Please check p2, 2.2 Cell culture: first paragraph.
Comment 3: In Figure 1, the authors showed the increased expression of TRPV2 proteins in HG group. The authors indicate which TRPV subtypes are expressed in mammalian lens epithelial cells and if their expressions are changes under HG treatments.
Answer: Thank you very much for your question. It has been reported that TRPV1 and TRPV4 are expressed in lens epithelial cells of pigs and mice [1-2], but we do not find the study of TRPV family in human lens epithelial cells in literature. It is really useful to clarify all members of TRPV family, however, it will be a huge work. We will use genomics and proteomics to figure out all TRPV members in following study. Thank you for providing us a good study plan.
Reference:
- Delamere, N.A.; Mandal, A.; Shahidullah, M. The Significance of TRPV4 Channels and Hemichannels in the Lens and Ciliary Epithelium. J Ocul Pharmacol Ther. 2016, 32, 504-508; DOI: 10.1089/jop.2016.0054.
- Nakazawa, Y.; Donaldson, P.J.; Petrova, R.S. Verification and spatial mapping of TRPV1 and TRPV4 expression in the embryonic and adult mouse lens. Exp Eye Res. 2019, 186, 107707; DOI: 10.1016/j.exer.2019.107707.
Comment 4: The authors should clarify the mechanisms underlying increased expression of TRPV2 proteins by HG stimulation. Decreased TRPV2 protein degradation or increased TRPV2 transcription, etc.
Answer: Thank you very much for your questions. Yes, it is really important for the TRPV2 mechanism study. It has been reported that the expression of TRPV2 in microglia is correlated with nitric oxide synthase (iNOS), protein kinase G (PKG) and phosphoinositol-3-kinase (PI3K) [1]. In retinal pigment epithelial cells, cannabidiol (CBD) and insulin-like growth factor-1 (IGF-1) enhance TRPV2 expression through PI3K pathway [2]. However, the specific mechanism of increased TRPV2 protein expression under HG stimulation is still unclear, which will be the focus of our future research.
- Maksoud, M.J.E.; Tellios, V.; An, D.; Xiang, Y.Y.; Lu, W.Y. Nitric oxide upregulates microglia phagocytosis and increases transient receptor potential vanilloid type 2 channel expression on the plasma membrane. Glia. 2019, 67, 2294-2311; DOI: 10.1002/glia.23685.
- Reichhart, N.; Keckeis, S.; Fried, F.; Fels, G.; Strauss, O. Regulation of surface expression of TRPV2 channels in the retinal pigment epithelium. Graefes Arch Clin Exp Ophthalmol. 2015, 253, 865-74; DOI: 10.1007/s00417-014-2917-7.
Comment 5: In Figure 5, the authors suggested that HG-mediated (for 7 days) increase in ROS production might cause increased TRPV2 protein levels. To determine whether TRPV2 is upstream of ROS-dependent Bax/Bcl2 and caspase pathway, the authors showed the time-dependent changes in the TRPV2 and Bax/Bcl2 levels in Figure 3 and 5. Without the data, the reviewer cannot agree with the authors’ conclusion.
Answer: Thank you very much for your question. We understood that reviewer challenge us because we used 7 days HG treatment in Figure 3 experiments but used 3 days treatment in Figure 5 ROS experiments. To respond to this issue, we performed additional ROS assay experiments for 7 days HG treatment. Our results showed that ROS was significantly increased in HLEpiCs cultured in HG media for 7 days, but inhibited by an antioxidant Tempol (0.25, 0.5, 1 mM) treatment with a concentration-dependent manner. We added these data into new Figure 5b, please check this revised manuscript.
Comment 6: In cell models but not animal models of hyperglycemia (high glucose exposure), ion channel expressions are recovered after one week. Please explain the reasons why the HG-mediated increase in TRPV2 protein expression is maintained for over 7 days.
Answer: Thank you very much for your questions. In our previous study, HG-induced increase in Orai3 channel protein expression in HLEpiCs lasted for 14 days [1]. It was also reported that Orai1 protein expression increased in human glomerular mesangial cells (HMCs) after 7 days of high glucose exposure [2]. Sustained high glucose exposure above 7 days may have different effects on different ion channels. Our results showed that TRPV2 channel protein in HLEpiCs was still up-regulated after 7 days of high glucose exposure.
- Wang, Y.; Bai, S.; Zhang, R.; Xia, L.; Chen, L.; Guo, J.; Dai, F.; Du, J.; Shen, B. Orai3 exacerbates apoptosis of lens epithelial cells by disrupting Ca2+ homeostasis in diabetic cataract. Clin Transl Med. 2021, 11, e327; DOI: 10.1002/ctm2.327.
- Chaudhari, S.; Wu, P.; Wang, Y.; Ding, Y.; Yuan, J.; Begg, M.; Ma, R. High glucose and diabetes enhanced store-operated Ca(2+) entry and increased expression of its signaling proteins in mesangial cells. Am J Physiol Renal Physiol. 2014, 306, F1069-80. DOI: 10.1152/ajprenal.00463.2013.
Comment 7: In Figure 4, Ca2+ imaging experiments should be performed using TRPV2-knockdown cells.
Answer: Thank you very much for your question. It is yes. Maybe you missed some information in Figure 2. Actually, in Figure 2 (j,k), we have used specific siRNA to knock down TRPV2 protein expression in lens epithelial cells, and then conducted Ca2+ imaging experiments. Please take a look again.
Comment 8: English editing is needed.
Answer: Thank you very much for your advice. We have checked all language problems again in this revised manuscript. Please check this version of the manuscript.

Reviewer 3 Report
In this manuscript, the authors describe diabetic cataract and apoptosis of lens epithelial cells under high-glucose via transient receptor potential vanilloid 2 (TRPV2). They found that TRPV2 expression was enhanced in HLEpiCs and RLEpiCs and reactive oxygen species (ROS) produced by high glucose-loading induced the channel activity. Overall, experimental results are solid and the manuscript is well written. Following points should be addressed for further consideration.
- Materials and Methods (line 94); Why did the author use EGTA as a normal physiological saline solution? In this case, the authors should specify the free calcium concentration in the solution.
- Materials and Methods (line 173); The authors should clearly show where 2-way ANOVA was used.
- Results (line 187), "the expression of TRPV2 protein ..(Figure 1d-g)"; This explanation is confusing because the expression was not changed in RLEpiCs between 1 and 5 days.
- All figure legends; the explanation of sample numbers such as n=3-11 is confusing. Each sample number should be shown in each experiment.
Author Response
Point-to-point Response
General response: We kindly thank reviewer for your constructive suggestions. In response to your comments, we have followed your suggestions to revise the manuscript carefully. We believe that this version of the manuscript is largely improved.
#3 Reviewer
Comment 1: Materials and Methods (line 94); Why did the author use EGTA as a normal physiological saline solution? In this case, the authors should specify the free calcium concentration in the solution.
Answer: Thank you very much for your question. Sorry for our mistake. Actually, EGTA was not used in this experiment, and the free calcium concentration was 1 mM. We corrected this mistake. In some Ca2+-free solution experiment in question 2 of reviewer #1, we used 0.2 mM EGTA to chelate the extracellular Ca2+. Please check this revision of the manuscript.
Comment 2: Materials and Methods (line 173); The authors should clearly show where 2-way ANOVA was used.
Answer: Thank you very much for your questions. We added these information into this revised manuscript. Please check this version of the manuscript.
Comment 3: Results (line 187), "the expression of TRPV2 protein ..(Figure 1d-g)"; This explanation is confusing because the expression was not changed in RLEpiCs between 1 and 5 days.
Answer: Thank you very much for your questions. This was our writing problem. Actually the expression levels of TRPV2 protein in the two types of cells cultured in HG were significantly increased in the 7th day. We corrected it. Please check this version of the manuscript.
Comment 4: All figure legends; the explanation of sample numbers such as n=3-11 is confusing. Each sample number should be shown in each experiment.
Answer: Thank you very much for your questions. In Figure 1-4, the sample number of each experiment is shown in the images. The missed sample numbers in Figure 5 have been added. Please check this version of the manuscript.

Reviewer 4 Report
Linghui Chen showed that long-term hyperglycemia determined apoptosis of lens epithelial cells via TRPV2 channel overexpression. This is a novel information that should be corroborated by other experiments in order to render the manuscript suitable for publication in Cells.
Therefore, I suggest some experiments that could be easely addressed by the authors:
- Due to the importance of Ros production in this model, some other “mitochondrial markers” should be analyzed. For instance the authors may measure mitochondrial membrane potential or ATP production during overglucose exposure in epithelial cells.
- Figure 5 showed that TRPV2 is a ROS-regulated channel. However, no mention and comment about this important result is reported in the discussion's section.
- In figure 4 d caspase 3 is already expressed in the scrambled controls. Please provide experiments with the procaspase 3 expression or change the target, introducing another transductional element such as caspase 9 with a lower expression in the controls.
- The conclusion should report the statement on functional modulation of TRPV2 by oxidative stress
Author Response
Point-to-point Response
General response: We kindly thank reviewer for your constructive suggestions. In response to your comments, we have followed your suggestions to do additional experiments and revised the manuscript carefully. We believe that this version of the manuscript is largely improved.
#4 Reviewer:
Comment 1: Due to the importance of ROS production in this model, some other “mitochondrial markers” should be analyzed. For instance the authors may measure mitochondrial membrane potential or ATP production during overglucose exposure in epithelial cells.
Answer: Thank you very much for your advice. It is yes that there are several other parameters as mitochondrial markers. To respond to your concern, we did additional experiments. Mitochondrial membrane potentials of HLEpiCs under different conditions were measured after 7 days treatment. Our results suggested that the epithelial cell membrane potential was significantly decreased in high glucose environment, which was partially reversed by the transfection of TRPV2 siRNA or the treatment of cells with Tempol (1 mM) (Rebuttal Figure 4). In high glucose environment, increased apoptosis leads to the loss of mitochondrial membrane potential, and ROS-induced increase of TRPV2 may be involved in this process.
Method: After the treatment in different groups, HLEpiCs were loaded with fluorescent probe Rhodamine 123 (Beyotime, C2008S) diluted at a ratio of 1:1000 in serum-free medium and cultured at 37 °C with 5% CO2 for 30 min. Serum-free cell culture medium was used to wash excess fluorescent dye, and the results were examined by a fluorimeter.
Rebuttal Figure 4
Comment 2: Figure 5 showed that TRPV2 is a ROS-regulated channel. However, no mention and comment about this important result is reported in the discussion's section.
Answer: Thank you very much for your advice. We have added the discussion of this result. Please check this revised version of the manuscript.
Comment 3: In figure 4 d caspase 3 is already expressed in the scrambled controls. Please provide experiments with the procaspase 3 expression or change the target, introducing another transductional element such as caspase 9 with a lower expression in the controls.
Answer: Thank you very much for your advice. To respond to your concern, we performed additional experiments to introduce another transductional element caspase 9. Our results indicated that cleaved caspase-9 protein expression level in HG group was significantly increased compared with that in NG group. In the NG group, the cleaved caspase-9 protein expression level was not significantly altered between the HLEpiCs transfected with TRPV2 siRNA and scrambled siRNA. However, in the HG group, caspase-9 protein expression level was significantly decreased in TRPV2 siRNA transfected HLEpiCs compared with scrambled control siRNA transfected cells (Rebuttal Figure 5).
Rebuttal Figure 5
Comment 4: The conclusion should report the statement on functional modulation of TRPV2 by oxidative stress.
Answer: Thank you very much for your advice. We have added the description that oxidative stress regulates TRPV2 function in conclusion section. Please check this version of the manuscript.

Round 2
Reviewer 1 Report
We thank the authors for their responses and additional experiments performed. While the manuscript is overall improved, I still have some comments/questions.
Response to comment 2 :
An original trace must be provided in supplemental figure 2, in order to understand the protocol. The way it is described in the figure legend, it is not clear that 2-APB was applied in Ca2+ free medium followed by 1 mM Ca2+ re-addition.
Response to comment 3:
A trace must be provided to explain the protocol used: was it like for supplemental figure 2 or was 2-APB applied in Ca2+-containing medium. If this is the second protocol, then it is not appropriate to measure the amplitude of the response to assess the impact of BTP2 on Ca2+ entry.
According to the statistics shown in Fig S3, there is no effect of BTP2 under NG condition, on the contrary to what is stated in the text (line 273-275). This must be corrected.
Legend S3: “2-APB (250 μM) activated the channel with or without BTP2 (10 μM, an inhibitor of SOCE) treatment to induce the changes of intracellular Ca2+ concentration in normal physiological saline solution”. This sentence is not clear and should be rewritten.
Response to comment 4:
Point 1) References for 2-APB as activator of TRPV2 should be changed by the following:
H.Z. Hu, Q. Gu, C. Wang, C.K. Colton, J. Tang, M. Kinoshita-Kawada, L.Y. Lee, J.D. Wood, M.X. Zhu. 2-aminoethoxydiphenyl borate is a common activator of TRPV1, TRPV2, and TRPV3.
- Biol. Chem., 279 (34) (2004), pp. 35741-35748
- Juvin, A. Penna, J. Chemin, Y.L. Lin, F.A. Rassendren. Pharmacological characterization and molecular determinants of the activation of transient receptor potential V2 channel orthologs by 2-aminoethoxydiphenyl borate. Mol. Pharmacol., 72 (5) (2007), pp. 1258-1268
Point 3), the word “mainly” was not removed in the revised version of the manuscript.
Author Response
Point-to-point Response
General response: We kindly thank reviewers for their constructive suggestions. In response to reviewers' comments, we have followed their suggestions to do additional experiments and revised the manuscript accordingly. We believe that this version of the manuscript is improved again.
#1 Reviewer:
Response to comment 2:
An original trace must be provided in supplemental figure 2, in order to understand the protocol. The way it is described in the figure legend, it is not clear that 2-APB was applied in Ca2+ free medium followed by 1 mM Ca2+ re-addition.
Answer: Thank you very much for your comments. We added original traces into Supplemental Figure 2 to help understand the protocol. Please check the supplemental figures of this version.
Response to comment 3:
A trace must be provided to explain the protocol used: was it like for supplemental figure 2 or was 2-APB applied in Ca2+-containing medium. If this is the second protocol, then it is not appropriate to measure the amplitude of the response to assess the impact of BTP2 on Ca2+ entry.
Answer: Thank you very much for your questions. Yes. Our experiments may confuse readers to understand 2-APB-induced Ca2+ influx. To respond to this issue, we performed additional experiments. BTP2 is a nonselective inhibitor of Orai channels which have three types including Orai1, Orai2 and Orai3 mediating store-operated Ca2+ entry (SOCE). In the study, we used BTP2 to inhibit Orai channels. After HLEpiCs was cultured under NG or HG media for 7 days, cells were treated with BTP2 and then 2-APB induced Ca2+ influx in Ca2+-free solution containing (in mM) 140 NaCl, 5 KCl, 1 MgCl2, 10 glucose, 0.2 EGTA, and 5 HEPES (pH 7.4). Our results showed that in Ca2+ free solution, after treatment with BTP2 (10 μM), 2-APB (250 μM) induced an increase in intracellular Ca2+ concentration ([Ca2+]i) in HLEpiCs, and followed Ca2+ influx was evoked by application of 1 mM extracellular Ca2+. The increase of extracellular Ca2+ influx was enhanced in high glucose group (Supplemental Figure 2). This suggests that 2-APB-induced Ca2+ influx is increased in HG environment in HLEpiCs. We combined Supplemental Figure 2 and 3 into Supplemental Figure 2 and added this new data into Supplemental Figure 2. Please check again.
Supplemental Figure 2
(Please see the attached pdf)
According to the statistics shown in Fig S3, there is no effect of BTP2 under NG condition, on the contrary to what is stated in the text (line 273-275). This must be corrected.
Answer: Thank you very much for your careful check. Sorry for our mistake. We corrected this mistake. Please check this revision of the manuscript.
Legend S3: “2-APB (250 μM) activated the channel with or without BTP2 (10 μM, an inhibitor of SOCE) treatment to induce the changes of intracellular Ca2+ concentration in normal physiological saline solution”. This sentence is not clear and should be rewritten.
Answer: Thank you very much for your suggestion. We have replaced the original Figure S3. Please check the supplemental figures of this version.
Response to comment 4:
Point 1) References for 2-APB as activator of TRPV2 should be changed by the following:
H.Z. Hu, Q. Gu, C. Wang, C.K. Colton, J. Tang, M. Kinoshita-Kawada, L.Y. Lee, J.D. Wood, M.X. Zhu. 2-aminoethoxydiphenyl borate is a common activator of TRPV1, TRPV2, and TRPV3.
- Biol. Chem., 279 (34) (2004), pp. 35741-35748
- Juvin, A. Penna, J. Chemin, Y.L. Lin, F.A. Rassendren. Pharmacological characterization and molecular determinants of the activation of transient receptor potential V2 channel orthologs by 2-aminoethoxydiphenyl borate. Mol. Pharmacol., 72 (5) (2007), pp. 1258-1268
Answer: Thank you very much for your suggestions. We have added relevant references for 2-APB as an activator of TRPV2. Please check this revision of the manuscript.
Point 3), the word “mainly” was not removed in the revised version of the manuscript.
Answer: Thank you very much for your careful check. Sorry for our mistake. We corrected this mistake. Please check this revision of the manuscript.

Reviewer 2 Report
The authors addressed all comments and revised the manuscript accordingly. I have no more concerns.
Author Response
Point-to-point Response
#2 Reviewer:
The authors addressed all comments and revised the manuscript accordingly. I have no more concerns.
Answer: Thank you very much for your positive comment.

Reviewer 4 Report
This study address the role of TRPV2 in lens epithelial cells of diabetic patients using both human lens epithelial cell line (HLEpiC) and primary rat lens epithelial cells (RLEpiCs) cultured under high-glucose conditions.
The authors showed a detrimental role of this channels in the pathogenesis of catarat. In fact [Ca2+]i increase evoked by TRPV2 channel agonist was significantly enhanced in both HLEpiCs and RLEpiCs cultured in high-glucose media. The reduction of TRPV2 channel expression prevented apoptosis of these cells and could be considered a new avenue to follow against this pathology.
The manuscript is now suitable for publication
.
Author Response
Point-to-point Response
#4 Reviewer:
This study address the role of TRPV2 in lens epithelial cells of diabetic patients using both human lens epithelial cell line (HLEpiC) and primary rat lens epithelial cells (RLEpiCs) cultured under high-glucose conditions.
The authors showed a detrimental role of this channels in the pathogenesis of catarat. In fact [Ca2+]i increase evoked by TRPV2 channel agonist was significantly enhanced in both HLEpiCs and RLEpiCs cultured in high-glucose media. The reduction of TRPV2 channel expression prevented apoptosis of these cells and could be considered a new avenue to follow against this pathology.
The manuscript is now suitable for publication
Answer: Thank you very much for your positive comment.
